# Potential Clinical Applications of the Postbiotic Butyrate in Human Skin Diseases

**DOI:** 10.3390/molecules27061849

**Published:** 2022-03-12

**Authors:** Serena Coppola, Carmen Avagliano, Antonia Sacchi, Sonia Laneri, Antonio Calignano, Luana Voto, Anna Luzzetti, Roberto Berni Canani

**Affiliations:** 1Department of Translational Medical Science, University of Naples Federico II, Via Sergio Pansini 5, 80131 Naples, Italy; serenacoppola@mail.com (S.C.); lvoto4@gmail.com (L.V.); annaluzzetti@gmail.com (A.L.); 2ImmunoNutritionLab at the CEINGE-Biotecnologie Avanzate s.c.ar.l Research Center, University of Naples Federico II, 80131 Naples, Italy; 3Department of Pharmacy, University of Naples Federico II, Via Domenico Montesano 49, 80131 Napoli, Italy; carmen.avagliano@unina.it (C.A.); ansacchi@unina.it (A.S.); slaneri@unina.it (S.L.); calignan@unina.it (A.C.); 4European Laboratory for the Investigation of Food Induced Diseases (ELFID), University of Naples Federico II, 80131 Naples, Italy; 5Task Force on Microbiome Studies, University of Naples Federico II, 80131 Naples, Italy

**Keywords:** short chain fatty acids, skin microbiome, skin barrier, atopic dermatitis, psoriasis, UVB radiation, wound healing

## Abstract

Human skin is the largest organ and the most external interface between the environment and the body. Vast communities of viruses, bacteria, archaea, fungi, and mites, collectively named the skin microbiome (SM), cover the skin surface and connected structures. Skin-resident microorganisms contribute to the establishment of cutaneous homeostasis and can modulate host inflammatory responses. Imbalances in the SM structure and function (dysbiosis) are associated with several skin conditions. Therefore, novel target for the skincare field could be represented by strategies, which restore or preserve the SM natural/individual balance. Several of the beneficial effects exerted by the SM are aroused by the microbial metabolite butyrate. Since butyrate exerts a pivotal role in preserving skin health, it could be used as a postbiotic strategy for preventing or treating skin diseases. Herein, we describe and share perspectives of the potential clinical applications of therapeutic strategies using the postbiotic butyrate against human skin diseases.

## 1. Introduction

The skin is the largest human organ and represents the most external interface between the environment and the body [1]. The skin represents the first line of defense against infection, environmental stressors, and loss of nutrients and water, so addressing the skin is a gateway to overall health and well-being. Vast communities of viruses, bacteria, archaea, fungi, and mites, collectively named the skin microbiome (SM), cover the skin surface and connected structures (hair follicles, sebaceous glands, and sweat glands) [2]. The microbial composition changes across body sites, and shapes by their physical, chemical, and biological features such as anatomic location, local humidity, sebum and sweat production, host hormonal status, and age. Sequencing studies have shown that human SM includes approximately 113 phylotypes belonging to six bacterial divisions [3,4]. Most skin bacteria genera include *Staphylococcus*, *Propionibacterium, Micrococcus,* and *Corynebacterium.* Furthermore, the SM shares with the gut microbiome (GM), four main phyla: *Actinobacteria*, *Firmicutes*, *Proteobacteria,* and *Bacteroides* [4,5]. Skin-resident microbes contribute to the establishment of cutaneous homeostasis and can modulate host inflammatory responses [6]. Several of the beneficial effects exerted by the SM are aroused by the microbial metabolites short-chain fatty acids (SCFAs), acetate, propionate, and butyrate [7,8]. These microbial end-products can acidify the pH and thus inhibit the growth of other microbes, they can stimulate keratinocyte-derived immune mediators on the host epithelium, and on immune cells in the dermis and epidermis [6,9]. Furthermore, these fatty acids can decrease epithelial permeability improving the barrier properties, can elicit eutrophic effect on the skin, and can suppress cutaneous inflammatory response [10]. The mutualistic relationship between microbial communities and the host is essential for the establishment of a well-controlled and balance needed for healthy skin. Hence, a disruption of the SM homeostasis (dysbiosis) is associated with several skin conditions, either pathological such as acne, dermatitis, allergies, or dandruff or non-pathological such as reactive, irritated, or dry skin [10]. Therefore, novel targets for dermatologists and in the skincare field could be represented by strategies that restore or preserve the SM natural/individual balance [11]. SM dysbiosis linked with skin disorders, could be treated “on-site” via various mechanisms: prebiotics, probiotics, synbiotics, and postbiotics. Among them, postbiotics are the most easily formulated into products, contrary to probiotics that pose issues in terms of formulation and packaging to ensure the microorganisms’ viability [12]. The definition of postbiotics is “any factor resulting from the metabolic activity of a probiotic or any released molecule capable of conferring beneficial effects to the host in a direct or indirect way” [13]. Recent years have seen a sharp increase in clinical investigations of postbiotics use in dermatology and cosmetology, and particularly the beneficial action of the postbiotic butyrate has been described [14]. The opportunity of manipulating the SM to address human skin condition has paved exciting new paths for therapy. The SCFA butyrate is emerged to play a pivotal role in influencing the predominance of definite cutaneous microbic profiles, which subsequently evoke skin immune defense mechanisms, by protecting against infection and ultraviolet radiation, and providing adequate nourishment to the cells of the skin [15].

Herein, we describe and share perspectives of the potential clinical applications of therapeutic strategies using the postbiotic butyrate as modulator of the skin response to diseases, as a protective agent against skin damage, and as an enhancer of specific therapies.

## 2. Butyrate in the “Gut–Skin Axis”

Growing evidence has demonstrated a bidirectional crosstalk between the gut and the skin, referred to as the “gut–skin axis”, linking gastrointestinal health to skin allostasis [16]. Therefore, it is unsurprising that gut disorders are often accompanied by cutaneous manifestations.

Among environmental factors, diet plays a pivotal role in shaping the GM, which in turn it is dependent on food metabolites for its survival and metabolism [17]. Healthy dietary pattern, such as a high-fiber diet is essential for the maintenance of a healthy GM. Indeed, indigestible fibers provide high rates of butyrogenesis, which satisfy the epithelial cells metabolic requirements and enter the blood stream to exert immunomodulatory and epigenetic effects on other body sites, including the skin [18]. Therefore, skin health is modulated by nutrition, and dietary modulation could also represent a useful avenue for skin damage protection [19]. The GM modulatory effect on systemic immunity could represent the mechanisms by which gut exerts its influence on skin homeostasis [16]. Indeed, aberrant GM seems to be a contributor of the physiopathology of many inflammatory skin disorders (Figure 1) [20,21]. For instance, GM dysbiosis increases epithelial permeability and the leaky gut barrier gains access to the bloodstream of detrimental intestinal microbes and toxins, which once accumulate in the skin can disrupt its homeostasis impairing epidermal differentiation and barrier integrity [16]. GM dysbiosis determine the effector T cells activation, disrupting their balance with regulatory T cells (Tregs), the immunosuppressive counterpart [22]. The release into circulation of effector T cells and their pro-inflammatory cytokines are supposed to contribute directly to several skin dermatoses pathogenesis [22]. In turn, pro-inflammatory cytokines further increase gut permeability setting up a vicious cycle of systemic inflammation with deleterious consequences for the skin (Figure 2) [16,23].

The mechanisms responsible for the communication between the commensal bacteria of the skin and the immune system may be compared to what happens in the intestine [24]. Gut commensal microbes influence the mucosal immune system through the increase of Tregs, and this is mediated by the SCFAs [25]. Because skin commensal bacteria also contain SCFAs producing strains (e.g., *Cutibacterium acnes*), an immunomodulatory/anti-inflammatory mechanism like that in the gut also exists in the skin [26]. Most beneficial roles of SCFAs are mediated by direct activation of its G-protein coupled receptors (GPRs) GPR41, GPR43, and GPR109a, and by the inhibition of histone deacetylase (HDAC) [27]. Among SCFAs, butyrate suppresses immune responses by inhibiting cytokines and inflammatory cells production, and through the HDAC inhibition promotes Tregs proliferation, the main cells involved in many physiologic functions of the skin, such as hair follicle regulation, stem cell differentiation, and wound healing [28,29]. Inflammatory skin diseases are characterized by skewed cutaneous immune response and SM perturbation. Through the production of butyrate, commensal skin microbes may counteract exaggerated inflammatory responses by exerting a down-regulatory function and maintaining a homeostatic state under physiologic conditions [24]. Whereby, topical butyrate administration may become a useful therapeutic application with a “curative” potential on inflammatory skin diseases.

Figure 3 graphically describes the main effects of butyrate on the regulation of host functions.

In the next paragraphs, the beneficial effects of butyrate against the most common skin diseases have been reported.

## 3. Butyrate in Psoriasis Disease

Psoriasis is a chronic immune-mediated inflammatory skin disease, affecting more than 125 million people globally [30]. This condition is typified by enhanced tumor necrosis factor-α (TNF-α)/interleukin-23 (IL-23)/IL-17 axis, with hyperproliferating epidermal keratinocytes, and with anomalous differentiation [31,32]. Dendritic cells (DCs) secrete TNF-α, which acts on themselves and induces the secretion of IL-23, which in turn induces the conversion of Tregs into type 17 helper T (Th17) cells, which proliferate and overproduce IL-17A. IL-17A reduces forkhead box protein 3 (Foxp3) expression, suppressing Tregs functional activity and stability [31]. Indeed, functional defects in CD4+CD25+ Foxp3 Tregs affect patients with psoriasis, the main suppressors of the excess immune response and mediators of homeostasis. Defects in Tregs may contribute to psoriasis disease development and exacerbation [33]. The metabolites SCFAs of specific gut microbic populations (e.g., *Bacteroides fragilis*, *Faecalibacterium prausnitzii*, *Clostridium cluster* VI and XIVa) influence Tregs activity and number [34,35], and several pieces of evidence reported that psoriatic patients show a decrease in the GM abundance of protective taxa producing butyrate, which may contribute to the defects in Tregs, such as *Parabacteroides* and *Coprobacillus* [36], *Prevotella* and *Ruminococcus* [37], *Akkermansia muciniphila* [38], and *Faecalibacterium prausnitzii* [39]. Furthermore, it has been shown that the gut microbial genes encoding the enzymes involved in butyrate synthesis, butyrate kinase, and phosphate butyryltransferase are less abundant in psoriatic patients compared to matched non-psoriatic controls [40]. 

Acetylation of H3 histones is associated with the activity of Tregs, and it has been described that H3 acetylation is significantly decreased in Tregs of patients with psoriasis, compared to healthy controls [41]. Since butyrate acts on DCs to promote Foxp3 expression in Tregs, and it has a well-known role as an HDAC inhibitor, it has been demonstrated that it induces on Foxp3 intronic enhancer the histone H3 acetylation, allowing the expression of Foxp3 in naïve CD4+ T cells, inducing their differentiation into Tregs [42].

The impaired number and activity of Tregs in psoriasis determine deleterious effects on the ability to control the inflammatory response [43,44]. It has been shown that Tregs isolated from the blood of psoriatic patients have an altered suppressive activity, which was normalized through the topical application of sodium butyrate on human biopsies of psoriatic lesions [41]. Sodium butyrate topically applied normalizes the enhanced expression of IL-17 and IL-6 and restores IL-10 and FOXP3-expression levels [41]. Moreover, in the same study it has been demonstrated that sodium butyrate, though only topically applied, reduced also systemic inflammation response, since it was able to reduce splenomegaly and IL-17 expression and to induce IL-10 and Foxp3 in the spleen [41].

A more decreased expression of keratinocytes in psoriatic patients than in healthy controls of the butyrate binding receptors GPR43 and GPR109a [45] has been described. The topical appliance of sodium butyrate was able to increase the reduced expression of both receptors and was able to restore the altered cytokine balance in psoriasis via GPRs. Butyrate topical application caused an increase in IL-10 and IL-18 production, and a reduction in the cytokines, which block the suppressive activation of Tregs, IL-17, and IL-6 [45].

Altogether, this evidence indicates that the restoration of defective Tregs represents a promising therapeutic target for psoriasis disease. As stated, butyrate restores the defected Tregs, and it may represent a promising tool in the management of psoriasis.

## 4. Butyrate in Atopic Dermatitis

Atopic Dermatitis (AD) is a chronic inflammatory skin disease that usually begins in early infancy, but also affects a significant number of adults [46]. The skin damage in AD patients is caused primarily by chronic inflammation, high levels of immunoglobulin (Ig)-E in the serum and anomalous T helper (Th)-2 type immune responses, with an overproduction of pro-inflammatory cytokines against common environmental triggers [47,48,49]. The GM, regulating the immune system development and function, might play a crucial role in AD. Indeed, imbalance in GM composition (dysbiosis) and a decreased production of SCFAs have been reported to precede the AD onset [50,51,52,53]. Low fecal levels of SCFAs have been linked to AD development in infants [54], and higher level of gut butyrate-producing bacteria have been reported in healthy infants than in those with severe AD [55,56,57]. It has been demonstrated that AD patients have an imbalance of *Faecalibacterium prausnitzii* subspecies compared to healthy subjects that results in a significant reduction in butyrate production, with a suppression of other subspecies butyrate-producing, such as the A2-165 type bacteria, resulting in gut barrier damage through an increase of various pathobionts [58]. This condition determines the entry into the systemic circulation of toxins and pathogenic microbes that can finally reach the skin and induce aberrant Th2-type immune responses to allergens. As stated above, butyrate exerts anti-inflammatory effects, preserving tight junctions and mucus layer, and promoting Tregs formation to increase IL-10 production, protecting against AD occurrence [59,60]. In AD murine model, the oral intake of a probiotics mixture plus sodium butyrate determined an increase of Th1 and Tregs differentiation, and of the population of butyrate-producing bacteria, thereby alleviating AD symptoms [61]. Furthermore, butyrate in addition to the anti-inflammatory effects, also exerts *Staphylococcus aureus* bactericidal activity. As is known, the skin of AD patients is more susceptible to the *Staphylococcus aureus* colonization and overgrowth [62]. Glycerol fermentation of *Staphylococcus*
*epidermidis* determined the production of butyric acid and effectively hindered the *Staphylococcus*
*aureus* strain growth in skin lesions of AD patients, in vitro and in vivo [63], confirming the immunomodulatory effects of butyrate in mitigating AD.

## 5. Butyrate Effects against Skin Prolonged Exposure to Ultraviolet B (UVB) Radiation

The skin and its constituents represent the most external layer of the body, and as such, are the primary targets for solar Ultraviolet B (UVB) radiation [64]. The chronic exposure to UVB radiation alters the cutaneous and systemic immune systems, causing several skin cells signaling alterations resulting in erythema, sunburn, inflammation, and carcinogenesis [65]. UVB radiation promotes activation of specific receptors on cell surface activating biological process such as DNA damage, lipid peroxidation, generation of reactive oxygen species (ROS) and chronic inflammation due to the release of key inflammatory mediators such as IL-1, -6, -8, -10, and TNF-α [66]. The effect of a pre-treatment with sodium butyrate in human fibroblasts have been previously investigated. Butyrate induces changes of cellular nucleotide metabolism and stimulates repair of UV damage by increasing the rate of specific DNA deletion modifying the structure of chromatin [67]. Furthermore, butyrate induces hyperacetylation of histones H3 and H4, facilitating the access of DNA repair enzymes to damage sites [68,69]. This process influences the average number of nucleotides incorporated at each repair site, the expression of certain proteins involved in repair synthesis after UV damage, and the rate of removal of UV-induced lesions after irradiation by excision [67].

The commensal bacteria of human skin *Staphylococcus epidermidis* can ferment the major component in stratum corneum glycerol, into butyric acid [70]. It has been shown that butyric acid alone or Staphylococcus epidermidis with glycerol topical application remarkably ameliorated the UVB-induced inflammation on mice skin [70]. Butyric acid noticeably decreased the ulceration of the skin and the epidermal thickness from UVB exposure and exerted a significant reduction in IL-6 and IL-8 level. These butyrate immunomodulatory effects are mediated by the binding with the GPR43, which controls the pro-inflammatory cytokines production elicited by skin injury [70]. Additionally, soothing and anti-reddening effects with a significant decrease of the erythema index induced by Sodium Laureth Sulfate (SLES) were demonstrated in vivo after the application of a butyrate releaser emulsion [14]. Therefore, butyrate can be effective not only for overt skin diseases, but as an ingredient in cosmetic products, to prevent skin alterations.

## 6. Butyrate in Skin Wound Healing

Skin homeostasis disruption may be associated with several cutaneous diseases and abnormal skin wound healing [71]. Wound healing process is based on a precious molecular mechanism divided into three different steps such as inflammation, cell proliferation, and cell differentiation [72]. The healing process can be hampered in the case of wounds that are large, long-lasting, and difficult-to-treat. Given the wound healing process complexity, the development of functional wound dressing materials that stimulate reparative and regenerative processes and have a positive effect on infected and/or difficult-to-heal wounds are needed [73]. The effects of a porous dressing materials based on butyric-acetic chitin co-polyester containing 90% of butyryl and 10% of acetyl groups (BAC 90/10) have been evaluated [74]. In vivo results have shown that BAC 90/10 had beneficial effect on skin repair and epithelization process, evoked reduced inflammatory effect with less effusion, and enhanced the creation phase [74]. Furthermore, the effects of sodium butyrate in combination with two different growth factors (EGF and PDGF-BB) in the skin wound healing of diabetic mice [75] have been reported. Butyrate plays a role as cell differentiation factor, an essential process for a normal diabetic wound healing. The combination of growth factors with sodium butyrate could be a new therapeutic agent as nanoparticle to treatment of human diabetic wounds [76]. Furthermore, a treatment by 4-phenylbutyrate (4-PBA) of venous leg ulcer biopsies showed a reduction of endoplasmic reticulum (ER) stress markers [77]. In effect, several cellular stress conditions are associated to accumulation of unfolded proteins in ER lumen, by a process called ‘ER stress’ [78]. This mechanism is overcome in cells by an unfolded protein response (UPR), a mechanism typical of eukaryotic organisms to bypass cellular damage induced by ER stress. In the case of a venous ulcer, a cellular stress conditions as hypoxia, induces an unfolded or misfolded of different proteins accumulate in the ER, causing ER stress [79]. Typical progress of wound healing is divided into inflammatory, proliferative (neo angiogenesis, tissue formation, re-epithelization), and tissue remodeling phases. In biopsies of venous ulcers, a treatment by 4-PBA increased the rate of re-epithelization, suggesting a potential use of this compound in wound healing therapeutic products [80].

## 7. Conclusions

Effective strategies for preventing and treating skin disorders could be the development of approaches that preserve or restore the SM natural/individual balance. Emerging data are supporting the efficacy of the topical application of the postbiotic butyrate. This strategy is considered safe, free from toxic and side-effects, industrially scalable, and easily formulated into cosmetic products. Unfortunately, the main limitation factor for the use of butyrate in dermatology could be due to the unfavorable sensorial and physicochemical properties. Nevertheless, products that mask the unpleasant organoleptic properties of butyrate are needed to facilitate its use in clinical practice. Odorless butyrate releasers could be an effective strategy for facilitating a wide use of butyrate against human skin disorders.

## Figures and Tables

**Figure 1 molecules-27-01849-f001:**
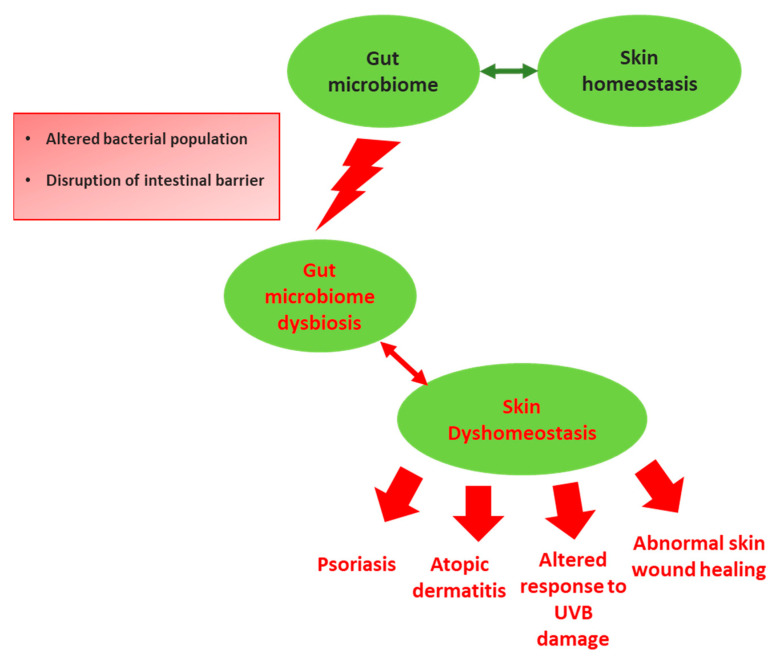
Schematic illustration of the correlation between gut microbiome (GM) and skin homeostasis. GM is the major regulator of the gut–skin axis: in fact, perturbations of GM homeostasis (dysbiosis) provoke also an altered skin environment. This condition increases the predisposition for the host to develop skin diseases and/or altered responses to skin damages.

**Figure 2 molecules-27-01849-f002:**
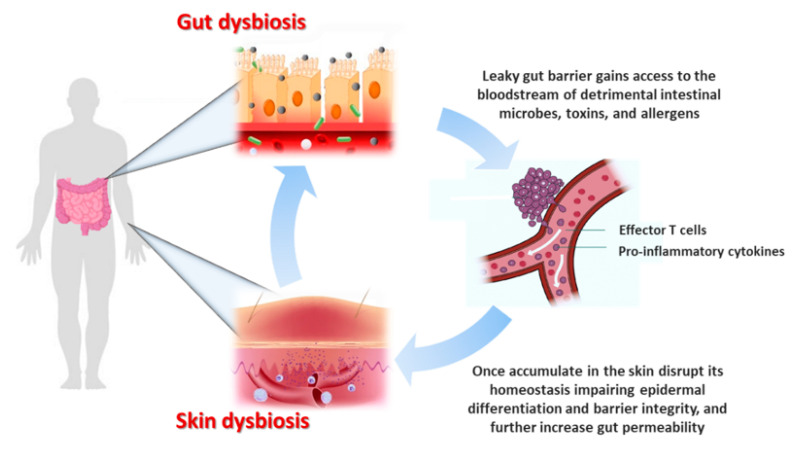
Given the bidirectional crosstalk between the gut and the skin, referred to as the “gut–skin axis”, it is unsurprising that gut disorders are often accompanied by cutaneous manifestations. GM dysbiosis increases epithelial permeability and the leaky gut barrier gains access to the bloodstream of detrimental intestinal microbes and toxins, which once accumulated in the skin can disrupt its homeostasis impairing epidermal differentiation and barrier integrity. GM dysbiosis trigger the activation of effector T cells, disrupting their balance with immunosuppressive counterpart regulatory T cells (Tregs). These effector cells and their pro-inflammatory cytokines are supposed to directly contribute to the pathogenesis of several skin inflammatory dermatoses. In turn, pro-inflammatory cytokines further increase gut permeability setting up a vicious cycle of systemic inflammation with deleterious consequences for the skin.

**Figure 3 molecules-27-01849-f003:**
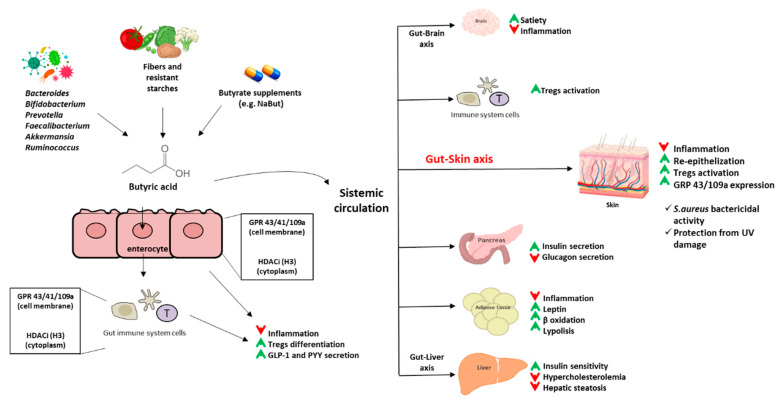
Schematic illustration of the main effects of butyrate on the regulation of host functions. Butyrate positive effects are indicated with a green arrow, negative ones are indicated with red arrows. Abbreviations: NaBut: sodium butyrate; GPR: G-protein coupled receptor; HDACi: histone deacetylase inhibitor; Tregs: T regulatory cells; GLP-1: Glucagon-like peptide 1; PYY: Peptide YY.

## Data Availability

No new data were created or analyzed in this study. Data sharing is not applicable to this article.

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
