# Peer review of "Potential Clinical Applications of the Postbiotic Butyrate in Human Skin Diseases"

_molecules, 2022, doi:10.3390/molecules27061849_

Round 1

Reviewer 1 Report

The paper entitled “Potential Clinical Applications of the Postbiotic Butyrate in Human Skin Diseases” provides an interesting literature review of studies about the beneficial effects of using the postbiotic butyrate for skin diseases treatment. The manuscript is well written and deals with the main aspects of the topic, however I believe they could get some improvement by working on the following key points:

  • It would be interesting to mention the main bacteria belonging to the skin microbiome (are they the same as in the gut microbiome?);
  • Due to the importance of the gut-skin axis, as highlighted by the authors, it would worth to mention the role of diet or natural compounds in maintaining a healthy gut microbiome, and/or a healthy skin. I can suggest some interesting readings regarding this topic: https://doi.org/10.1186/s12967-017-1175-y, https://doi.org/10.1007/s10620-020-06112-w, https://doi.org/10.1016/j.humic.2019.100063, https://doi.org/10.1146/annurev-food-032519-051722, https://doi.org/10.3390/md19070379 ;
  • Including more figures would be appreciated by the readers of the review. For example, a figure representing the main effects of skin dysbiosis (before the one showing the crosstalk with the gut dysbiosis), and another regarding the protective effects of butyrate (including like its formula, biosynthetic pathway, the microbiome bacteria species producing it, and effects of this molecule at molecular and systemic level).

Minor issues:

  • The first sentence of the abstract is exactly the same as the first one in the manuscript, please re-phrase one of the two;
  • Mention abbreviations the first time of appearance in the text (example: Treg).
  • The authors discuss about the bidirectional crosstalk happening between the gut and the skin, but the figure 1 shows directional talk instead, please represent it better.

Author Response

In the lines 39-43 we have mentioned the main bacteria belonging to the skin microbiome; in the lines 81-88 we have mentioned the role of diet or natural compounds in maintaining a healthy gut microbiome, and/or a healthy skin; we have added a Figure representing the main effects of skin dysbiosis (Figure 1), and another regarding the protective effects of butyrate (including like its formula, biosynthetic pathway, the microbiome bacteria species producing it, and effects of this molecule at molecular and systemic level) (Figure 3); we have rephrased the first one sentence of the manuscript; we have mentioned abbreviations the first time of appearance in the text; and we have represented better the bidirectional crosstalk happening between the gut and the skin in the Figure 2.

Reviewer 2 Report

In this review manuscript, the authors describe and share perspectives of the potential clinical applications of therapeutic strategies using the postbiotic butyrate against human skin diseases, such as psoriasis, atopic dermatitis, skin wound healing, et al. Overall, too many text descriptions make it difficult to understand the mechanism of butyrate’s effects on skin health. It will be better that the authors can add some simple diagrams or graphs to show the signaling pathways of postbiotic butyrate. In addition, the mechanism of the effects of butyrate on skin seems to be still not fully clear yet. So, one of the key points in the future is to clarify the mechanism, especially in human and on this basis, the development of butyrate products for the treatment of skin diseases should be targeted.

Author Response

We have added a Figure (Figure 3) to represent graphically the mechanism of butyrate’s effects on skin health and on host functions, showing its signaling pathways.